# Analytical Evaluation of Carbamate and Organophosphate Pesticides in Human and Environmental Matrices: A Review

**DOI:** 10.3390/molecules27030618

**Published:** 2022-01-18

**Authors:** Nonkululeko Landy Mdeni, Abiodun Olagoke Adeniji, Anthony Ifeanyi Okoh, Omobola Oluranti Okoh

**Affiliations:** 1Department of Chemistry, University of Fort Hare, Alice 5700, South Africa; ookoh@ufh.ac.za; 2SAMRC Microbial Water Quality Monitoring Centre, University of Fort Hare, Alice 5700, South Africa; adenijigoke@gmail.com (A.O.A.); aokoh@ufh.ac.za (A.I.O.); 3Department of Chemistry and Chemical Technology, Faculty of Science and Technology, National University of Lesotho, Roma 180, Lesotho; 4Department of Environmental Health Sciences, College of Health Sciences, University of Sharjah, Sharjah P.O. Box 32223, United Arab Emirates

**Keywords:** pesticides, carbamates, organophosphates, toxicity, bioavailability

## Abstract

Pesticides are synthetic compounds that may become environmental contaminants through their use and application. The high productivity achieved in the agricultural industry can be credited to the use and application of pesticides by way of pest and insect control. As much as pesticides have a positive impact on the agricultural industry, some disadvantages come with their application in the environment because they are intentionally toxic, and this is more towards non-target organisms. They are grouped into chlorophenols, organochlorines, synthetic pyrethroid, carbamates, and organophosphorus based on their structure. The symptoms of exposure to carbamate (CM) and organophosphates (OP) are similar, although poisoning from CM is of a shorter duration. The analytical evaluation of carbamate and organophosphate pesticides in human and environmental matrices are reviewed using suitable extraction and analytical methods.

## 1. Introduction

Pesticides and insecticides are a group of compounds that are toxic to pests and insecticides, respectively. The application of insecticides is advantageous to agriculture and agriculture and for the extirpation of vectors [1]. Pesticides are used in the agricultural industry to control pests, and it is because of this that there is such high productivity achieved in this industry. They are intentionally released into the environment, and this comes with adverse effects in the environment, because they are toxic and often affect non-target organisms [2]. Pesticides can be categorized into organochlorines, organophosphorus, carbamates, chlorophenols, and synthetic pyrethroids based on their structures [3]. Carbamates are a group of insecticides that are similar to organophosphate pesticides both structurally and mechanistically. Carbamates are N-methyl carbamates that originate from amino formic acid. The difference between carbamates and organophosphates is that carbamates bind to acetylcholinesterase reversibly, while the phosphorylation of acetylcholinesterase by organophosphates is irreversible [4]. Some commonly used carbamate pesticides include carbaryl, carbofuran, and aminocarb (Figure 1) [5].

Esters derived from the simplest carbamic acids are generally unstable compounds, particularly in basic conditions. The ester derivatives of carbamates are crystals with a low vapour pressure (Pv) and low but varying water solubility. They have poor solubility in nonpolar organic solvents such as chloroform and toluene, and, in contrast, are highly soluble in acetone, a polar organic solvent [6].

OPs are commonly used in agriculture, but some, such as malathion, treat hominids with scabies, head lice, and crab lice [7]. OPs are also used in farm cultivation, veterinary medicine, and public hygiene to control paths (routes) of diseases. Organophosphorus pesticides inhibit cholinesterase, so they are generally more deadly to invertebrates. The acetylcholine neurotransmitter permanently overlaps across a synapse, causing muscle trembling, which leads to paralysis and can eventually be fatal. Unlike OCs (organochlorines), OPs are short-lived in the environment; that is, they are not pertinacious. Some common organophosphates are glyphosate, parathion, diazinon, and malathion, as shown in Figure 2 [5]. 

Carbamates (CMs) and OPs are strong choline esterase inhibitors and can lead to cholinergic poisoning if inhaled, ingested, or exposed to the skin. For more than five decades, organophosphates have been used as insecticides worldwide. However, their application has decreased in the last two decades or so due to the expansion of carbamate insecticides, which are linked with similar toxicities [8]. Alzheimer, the treatment of glaucoma, and the reversal of neuromuscular blockade are all within the medical application of organophosphate and carbamates. In the 1930s, organophosphorus (OP) and carbamate compounds were used as pesticides and were not persistent in the environment. As temperature, pH, or both increase, the chemical breakdown of these pesticides accelerates, and their toxicity is due to the disruption of the nervous system of an invertebrate or a vertebrate through the inhabitation of cholinesterase (ChE) enzymes [9].

Carbamate and organophosphate pesticides from industrial wastes, spills, accidental releases, and illegal dumping may enter rivers, creeks, and wetlands; thus, the aquatic environment is vital in the transportation of pesticides. However, because they decompose in water, their danger is limited. Even though long-term contamination by these pesticides is unlikely, marine animals may be harmed due to run-off after application [10]. These compounds are widely used in farming, forestry, parks, and educational facilities to control pests and insects that can act as disease vectors by repelling, preventing, mitigating, and destroying the insects and weeds. Pesticides can be grouped based on their modes of action into fungicides, rodenticides, insecticides, wood preservatives, molluscicides, herbicides, and bactericides [3]. The different types of pesticides and their targets are shown in Table 1.

Pesticides have become remarkably helpful in protecting plants from insects, pests, and diseases; this has resulted in pollution, in turn, causing increasing worry by the public. The application of most hydrophobic organic pesticides is limited due to their low water solubility and their soil removal difficulties [12]. Carbamates are used excessively in countries whose major business is agriculture, where pest control is very crucial. They are regularly used in place of OP insecticides [7]. Table 2 shows the estimated percentage losses of crops caused by pests per year. Therefore, it is obvious that the uncovering of pesticides has not been a luxury of a technical civilization but somewhat a need for the survival of humankind [5].

Organophosphate pesticides were developed in the early 19th century; however, that their effect on insects is comparable to their impact on humans was discovered in 1932. They were used as nerve agents in World War II, owing to their toxicity. Carbamates are progressively replacing OP and OC pesticides because their half-lives are shorter in the environment. However, since they inhibit choline esterase, CMs are suspected to be carcinogens and mutagens. Therefore, gradual usage of carbamate pesticides can potentially be hazardous to the human environment [13]. 

## 2. Physicochemical Properties and Applications of Carbamate and Organophosphate Pesticides

### 2.1. Physicochemical Properties

Pesticides have become a problem globally because of some elements that may influence their transport and destiny. Understanding these elements is essential to understand why pesticides are a global hazard. To establish how pesticides are distributed between air, soil/sediments, water, and organisms, simple physicochemical features of pesticides may be used [14]. The three partition coefficients, K_AW_ (air/water), K_OW_ (octanol/water), and K_OA_ (octanol/air), together with the solubilities of chemicals in the liquid phase (S_A_, S_W_, S_O_; expressed in mol m^−3^ in water, octanol, and air) are the major properties describing the partitioning of phases in the environment. Hydrophobicity is measured by K_OW_, which is also a correlation property in bioaccumulation assessment and is used for partitioning in sediment, organic carbon, and water. K_OA_, S_A,_ or liquid-phase vapour pressure (P_L_ = S_A_RT, Pa) are all correlation properties used to describe organic compounds that are adsorbed to aerosols. K_OA_ has further applications in modelling soil–air exchange and bioaccumulation through the respiratory exchange. K_H_ (Henry’s law constant) is used in the estimation of the direction and rate of air–water gas exchange and precipitation scavenging [15].

Factors such as temperature strongly affect water solubility, P_v_ (vapour pressure), and K_H_ (Henry’s law constant). In the summer season, volatilization by the warming of surface water is higher due to high temperatures. The solubilities of OP and CM pesticides differ from solvent to solvent; for example, they are more soluble in organic solvents than in water [14].

#### 2.1.1. Organophosphate Pesticides

OPs are originally synthetic and usually are amides, esters, or thiol derivatives of phosphoric or phosphonic acids. Figure 3 below shows the general structure of organophosphate pesticides.

Most OPs have low vapor pressure, a high oil–water partition coefficient, low volatility (except dichlorvos), and are moderately soluble in water. Some organophosphorus pesticides, such as parathion, chlorpyrifos, phosalone, and diazinon, are very lipophilic and can remain in a human body for days or weeks in severe cases. Chlorpyrifos (CPY), for example, is a hydrophobic compound that links strongly to sediments once it enters the water. This insecticide is also relatively persistent in sediments, with a half-life of 30 days [16]. Its vapor pressure is 1.73 × 10^−1^ torr, its solubility in water is <1 mg/L, and its log K_OW_ is 5. The mean water–soil adsorption coefficient, normalized to a fraction of the organic carbon in the soil (K_OC_), of CPY is 8.2 × 10^3^ mL g^−1^. The half-lives of CPYs in soils are in the range of 2 to 1575 d (*n* = 126) in laboratory conditions, depending on properties of the soil and rate of application [17]. The way a pesticide will react once it reaches the soil will depend on its properties such as size, structure, functional groups, and polarity of the molecule, including the resulting dissociation constants and partitioning coefficients (e.g., Ka, and Koc) therefrom. This is why different soil types have different application rates [18]. Table 3, below, shows the physical and chemical properties of OP pesticides.

The fact that soils and sediments can bind different chemicals to varying degrees, suggests bringing national attention to bioavailability. The availability of these chemicals to surface water, groundwater, air, and all living organisms is therefore altered. The physiological features of plants and animals impact the availability of chemicals, meaning that the consequences of contact with the same contaminated material will differ from one species to another [18]. Table 4 shows the half-lives of some OP pesticides. 

OPs concentration was detected in environmental matrices, such as snow, air, and rain, from locations far from agricultural sources, suggesting that long-range transport (LRT) is a possibility [20]. Pesticides can also be spread to the atmosphere (in air) when they are absorbed in the particles of soil that are blown by the wind. The vaporization of pesticides applied to the surfaces of plants has proven to be significant; for instance, laboratory results from chamber experiments revealed substantial volatilization percentages (60%, 40%, 50–80%, 40–70%) for endosulfan, parathion-ethyl, parathion-methyl, and fenpropimorph, respectively. Factors influencing the spread of substances include vertical transport to higher layers, seasonal distribution, land–sea differences, and exchange and removal processes. The time for the spread of pesticides through long-range transport can be anything from hours to a couple of days; by then, the pesticides have formed homogeneous mixtures in the atmosphere [23].

#### 2.1.2. Carbamate Pesticides

CM pesticides are similar to OP pesticides in their mechanistic and structural forms (Figure 4) [4].

R^2^ is an aromatic or aliphatic moiety. The three main classes of carbamate pesticides are known: carbamate insecticides, where R^1^ is a methyl group; carbamate herbicides, where R^1^ is an aromatic moiety; and carbamate fungicides, where R^1^ is a benzimidazole moiety [6]. The relationship between pesticidal activities and the chemical structures of some carbamate pesticides are shown in Table 5.

A knowledge of the acid–base ionization properties of organic molecules is essential to describing their environmental transport and transformations or estimating their potential environmental effects. For ionizable compounds, solubility, partitioning phenomena, and chemical reactivity are all highly dependent on the state of ionization in any condensed phase. The ionization pKa of an organic compound is a vital piece of information in environmental exposure assessment. It can be used to define the degree of ionization and resulting propensity for sorption to soil and sediment; consequently, it can determine a compound’s mobility, reaction kinetics, bioavailability, and complexation [26]. The chemical identities and a summary of the chemical and physical properties of some carbamates are listed in Table 6 below [6].

Organophosphates and carbamates are volatile compounds following the classification of Woodrow and Seiber (1983) and Unsworth et al. (1999) [23]. There are two ways that the volatility of a compound can affect its fortune in the environment. The first one is the control of compounds’ partitioning between the particle phase and the vapour phase, and the second way is by controlling the partitioning of compounds between the dissolved phase in water and the vapour phase in the atmosphere. These are controlled by Henry’s law constant (water solubility) and vapour pressure [28]. Volatility is one of the physicochemical properties considered when assessing exposure pathways; others include solubility, boiling and melting points, and K_OW_ (log*P*) to name a few [29]. 

The atmospheric long-range transport of some pesticides that have been used in the past has been recorded. Their degradation and occurrence in locations far from application sites have been documented by data monitoring. As helpful as data monitoring is in the documentation of long-range transport, it has the disadvantage of not being able to predict the possibility of such transportation across long distances before it happens. The second demerit is that it is difficult to institute a relationship between far-field pesticide concentrations and near-field pesticide loadings using only data monitoring. These relationships are needed to evaluate the influence of risk mitigation options on potential long-range transport [30].

Whether or not a pesticide will be removed by surface water or rain from the atmosphere or if volatilization will occur from a surface is dependent on its vapour pressure and solubility. Carbamates are soluble in water, with varying vapour pressures. The balance of these two processes is represented by the Henry’s law constant, K_H_ [28]. K_H,_ together with K_OW_ are the properties that determine whether a chemical will be persistent in the environment; if K_H_ is large, that means volatilization of the pesticides will be favoured [31]. CMs found in sediment and soil are adsorbed to the OC fraction (organic carbon). The soil adsorption coefficient, K_d_, measures the movement of a substance in the soil. When a high value is obtained, it means that the substance is immobile in the soil and adsorbed strongly onto the organic matter in the soil, while a low value suggest that the substance is mobile. Koc is a vital input parameter for the estimation of environmental distribution and exposure level of a chemical substance [28]. K_AW_ and K_OW_ can be used to predict the transport of CM and OPs with low values indicating that long-range transport in the air is less likely [32]. From the partition coefficient (Kp), Koc can be derived. Kp is a ratio the concentration of a chemical substance associated with particulates to that in the solution [28]. Vapour pressure and solubility can influence K_AW_, which is the air–water partition coefficient. The soil–water partition coefficient of low-polarity chemicals and fish–water bioconcentration factors may be estimated from the organic content of the soil. In contrast, the organic carbon–water partition coefficient (Koc) may be assessed from the octanol–water partition coefficient (Kow). Amounts and concentrations of phases in equilibrium using, a mass balance constraint, can be calculated by directly using partition coefficients or the fugacity approach. OPs and CMs have different partition characteristics, mostly a result of their solubilities [33].

## 3. Sources of CM and OPs in the Environment

Carbamates and organophosphates are known to act through a similar mechanism of action. The pathways of exposure of OPs can overlap for many wildlife and humans; an example would be the drift of pesticides from sprayed crops to communities nearby. Equally, the run-off of pesticides pollutes the water, which can be harmful to marine organisms, land-dwelling species that live and feed around water bodies, and people in such location. Contamination by OPs can be detrimental to humans long after the adverse impact of pesticide spraying [34]. Several physical, biological and chemical processes can influence pesticides after they enter the environment. Once pesticides are in the environment, they leach into groundwater and can remain there for a long period. Degradation is slowed down because there is little or no light underground, hence, increasing the potential risks of pesticides to the health of humans and the environment. Some are eaten by organisms, while many are degraded by environmental and microbial processes [35]. Figure 5 below shows the routes of exposure of carbamate and organophosphate pesticides to wildlife and humans.

## 4. Toxicity of CM and OPs and Risk of Exposure

Most countries use pesticides in their defence, civilian, and commercial sectors for a diversity of purposes. That has resulted in vast organic pollution, which is now a global challenge [35]. Billions of kilograms of pesticides are used all over the world, with CMs and OPs (34%) being the most used every year; it has been reported that a gardener is likely to be exposed to pesticides, whether directly or indirectly [36]. 

Given their widespread use, CMs and OPs are considered environmental contaminants, as they can be a hazard to the health of humans by their accumulation in food and the environment [37]. CM and OP act by inhibiting the activity of acetylcholine esterase (AChE), which is essential for the central nervous system’s functioning in insects and humans. When acetylcholine esterase is not active, the acetylcholine neurotransmitter accumulates, and this can lead to the malfunction of respiration and muscle tissue of the heart and eventually lead to death [13]. CMs bind the AChE receptor reversibly, while OPs bind it irreversibly, so recovery from poisoning by OPs may need the synthesis of new enzymes, which may take up to a few weeks. On the contrary, recovery from CM poisoning is fast and only takes a few hours [38].

The assessment of chemicals for safety before being approved for use by humans is essential. Toxicity testing commenced in 1520 and has since been improved by considering replacement, refinement, and reduction, so new approaches have been developed. For a long time, the median lethal dose (LD50) test methods have been used for testing the acute toxicity of chemicals. However, they have recently begun being suspended because newer methods have been developed and authorized by regulatory bodies. Some of these new methods include FDP (fixed-dose procedure), UDP (up-and-down procedure) usually involving several animals, and the ATC (acute toxic class) method. All the methods mentioned have been approved based on refinement and reduction approaches. Other methods approved for the replacement approach include the NHK (normal human keratinocyte) and NRU (3T3 neutral red uptake), for acute phototoxicity tests. Other alternative methods have not yet been approved; this means that a united effort from the academic community and science organizations is needed to ensure their fast approval [39].

Carbamates are considered mutagenic and carcinogenic because they can be converted to N-nitroso compounds. However, the amount of these compounds resulting from the ingestion of CM pesticides is negligible compared to the precursors of nitroso occurring naturally in drinking water and food [6]. High levels of exposure to OP pesticides can be fatal in a short period, and a few studies have proposed that chronic low-level exposure, mainly for children, may lead to health complications. In a study by Boyd Barr et al. (2010), it was found that exposure to OP pesticides in early childhood can lead to disorders such as ADHD (attention deficit/hyperactivity) and neurological disorders with children residing in agricultural areas at a higher risk [40].

## 5. Sample Collection and Preservation

Sample collection and analysis are vital in any environmental assessment because their quality will only be as good as the data found through sampling. Environmental samples are collected in different ways, by various groups, and for other reasons. One of the most essential steps in sampling is having background information on the sampling type that will be conducted because the quality of environmental data can for a given site differ from one sampling project to another [41].

### 5.1. Aqueous Samples

Amber glass bottles are used as containers for sample collection following conventional sampling practices; field blanks are also collected to validate the sampling protocol. Freely flowing samples are collected using a sampling apparatus that is automated and as grab samples. If a high concentration of the pesticides is anticipated, volumes as high as 1 L of samples are collected. Residual chlorine that may be in the sample can be tested using methods suitable for field use and if found, 80 mg of thiosulfate is added to each sample for dechlorination. To avoid degradation of the samples, they must be kept in the dark at temperatures as low as 6 °C from the sampling site until further analysis. For samples that will be analysed 72 h after collection, H_2_SO_4_ or NaOH are used. A specific volume of base or acid is used to adjust the pH range to five to nine [42].

### 5.2. Solid, Semi-Solid, Mixed-Phase, and Oily Samples

Samples are collected as grab samples using wide-mouth sample containers. Wet materials are collected in adequate amounts to yield 20 g of solids. These solid, semi-solid, oily, and mixed-phase samples are kept in the dark at <6 °C immediately after collection until further analysis in the laboratory [43].

### 5.3. Fish and Other Tissue Samples

Fish samples are collected and cleaned, filleted, or even processed in different ways in the field to arrive in many forms in the laboratory (whole fish, fillets, or tissues). After the fish samples are collected, they are wrapped in aluminium foil and stored at <6 °C immediately after collection until further analysis with a maximum 24 h. If the samples may, for some reasons, be stored longer than 24 h, they are frozen under dry ice. Tissue samples are kept frozen and in the dark at −10 °C until use. The unused samples are left frozen and in the dark until analysis [42].

Preserving the samples until any analysis can be very important for obtaining the desired analytical results. Failing to do so may lead to many problems such as degradation of the samples. There must be no contamination, alteration, or loss of analytes during transportation, which can be a result of chemical or physical alterations in the stored samples. Collecting and preserving biological samples is an intricate problem, and this is correlated to the nature of the sample, the purpose of analysis, sample availability, among other factors [44].

## 6. Extraction Methods for CM and OPs in Water and Sediment

CM and OP pesticides are unstable. Due to this, their stable derivatives need to be prepared and indirectly analysed by GC or other techniques, since the pesticides concentration is too low to be detected directly in other samples. Thus, it is essential to implement previous pesticide enrichment and separation. Several pre-separation and pesticide enrichment techniques are reported in the literature, including their advantages and disadvantages. For pesticides in aqueous samples, LLE or liquid–liquid extraction is used, or alternatively solid-phase extraction (SPE), which uses less solvent and usually requires an extra step of concentrating the extract down to a small volume, is used and is commonly used for sample preparation [45]. Traditionally, solid samples are extracted mainly through a Soxhlet apparatus, but because of its disadvantages, such as being solvent- and time-consuming, it is being replaced by techniques that are more environmentally friendly and consistent with the current green chemistry analytical principles. Ultrasonic-assisted extraction (UAE) and pressurized liquid extraction (PLE) are the commonly used techniques that are based on new sources of energy. One of the advantages of PLE is the usage of little solvent, but it requires expensive, complicated equipment that consumes a lot of energy. At the same time, UAE is inexpensive with simple equipment [46]. Procedures such as microwave-assisted extraction (MAE) and supercritical fluid extraction (SFE) are used the least due to the purification step that is required before chromatographic analysis. Details of extraction methods for the pesticides in solid samples, including the advantages and disadvantages are provided in the Table 7 below.

## 7. Analytical Methods for CMs and OPs in Water and Sediment

Spectrophotometric and fluorometric methods were developed to help with the challenge of determining and identifying trace and ultra-trace pesticides. However, these methods are not highly specific, although they are sensitive. The ideal method for the analysis of pesticide residues should have high sensitivity, selectivity, accuracy, high precision, and low cost and should be applicable to a wide range of samples. Thus, several chromatographic techniques, such as high-performance liquid chromatography (HPLC), gas chromatography (GC), capillary electrophoresis (CE), and thin-layer chromatography (TLC), can be applied for their determination in different matrices [45]

### 7.1. Electrochemical Methods

Many electrochemical methods for pesticide analysis were reported. A cyclic voltammetry (CV) technique for quantifying pesticides was developed by Chen and Chen (2013), while square-wave voltammetry (SWV) with a boron-doped diamond electrode was used by Svorc et al. (2013) to detect nanomolar atrazine levels. A gold electrode surface electroplated with gold nanoparticles to immobilize the aptamer was used. When an acetamiprid–aptamer complex formed, a corresponding increase in the electron transfer resistance was correlated to nanomolar concentration levels of the insecticide [51].

In a study by [52], an electrochemical enzyme inhibition assay was used for the determination of carbamate and organophosphate pesticide (carbaryl and parathion methyl respectively) in milk, egg, honey, and bovine meat. The food samples were spiked with two different concentrations (10 and 30 ng/mL) of the pesticides of interest and tested. The assay allowed the detection of the tested pesticides at 10 ng/mL in the solvent extract of different complex matrices that were not purified, and the assay time was 15 min overall.

The modern electrochemical methods (amperometry and voltammetry), usually employed in the study and determination of pesticides in herbal medicine, crops, environmental and food samples, were reviewed by [53] some of the pesticides considered were carbamates, organochlorines and organophosphates. It was reported that electrochemical methods developed for the determination of some pesticides required a first step of pre-treatment by derivatization or hydrolysis. Few studies have used solid electrodes to investigate the electrochemical behaviour of pesticides directly on the surface of the electrodes. An analytical method that would use differential pulse-stripping voltammetry was proposed to determine zetran and aminocarb, with a detection limit of 30 mg L^−1^ for both compounds. The conclusion was that voltametric methods are beneficial due to their selectivity.

A nonenzyme electrochemical sensor for the determination of an organophosphate (methyl parathion) was established by Huo et al. (2018). In the analysis differential pulse voltammetry (DPV) technique was applied, where the linear range of MP (methyl parathion) concentration ranged from 1 ng/mL to 2 g/mL and the LOD was 0.53 ng/mL [54]. 

### 7.2. Spectroscopy

Surface-enhanced Raman spectroscopy (SERS) has been used for the detection of pesticides. The technique was used to detect the organophosphorus pesticide dimethoate. In the study, the method using confocal Raman micro-spectrometry with a klarite substrate was compared with the traditional Raman technique and it yielded significantly enhanced detection capabilities [55].

The removal of the organophosphate malathion in water using ultraviolet irradiation was investigated by [56] Shayeghi et al. (2012), where different concentrations of the pesticide were exposed to ultraviolet irradiation. It was found that the minimum reduction, which is 46%, happened at 10 min and 87.25%, which is the maximum reduction, occurs at 60 min. The dry-extract system for the (near) infrared technique was done using reflectance near-infrared spectroscopy to detect contact pesticide residues on mangos, apples, and tomatoes. It was concluded that the method would be best used only as a rapid screening tool [57].

### 7.3. Chromatographic or Mass-Spectrometric Techniques

Method 622 is an Environmental Protection Agency (EPA) method for the analysis of OP pesticides in both domestic and industrial wastewater using gas chromatography. It was designed to detect up to twenty-one pesticides in this class. OP compounds in the samples could be measured with flame photometric or thermionic bead detector in phosphorus mode. Further identification of analytes using GC–MS is encouraged, especially when dealing with unfamiliar samples. Interferences in this analysis could come from glassware, solvents, reagents, or any related apparatus used in the process. Hence, thorough cleaning is recommended before analysis. Similarly, the use of reagents and solvents with high degrees of purity cannot be over-emphasized. When analysing a very clean sample, the clean-up step may be skipped; however, where the purity of the sample is in doubt, appropriate clean-up procedures should be adopted. The dilution of sample extract is necessary if the instrument’s response for the analytes is higher than the working range of the instrument [58].

Method 1699 is another GC method for determining some classes of pesticides in different compartments of the environment. Among the classes of pesticides determinable with the method are the triazine, organochlorine, pyrethroid, and organo-phosphorus pesticides using high-resolution gas chromatography coupled with high-resolution mass spectrometry (HRGC–HRMS). Extraction of aqueous samples for this analysis at neutral pH could be done via separatory funnel or continuous liquid–liquid extraction method. Multi–phase, solid, semi solid, sewage sludge, fish and other tissue samples are otherwise extracted by means of Soxhlet extraction or a Soxhlet/Dean–Stark (SDS) extractor. For the GC determination, isotope dilution technique is used if a labelled analogue is available. However, if not available, then the use of internal standards is recommended. Interferences could come from the same as Method 622, earlier described, hence, proper cleaning of all apparatus and the procurement of reagents and solvents of best quality are encouraged [42]

Method 632 is an EPA-recommended method for determining carbamate and urea pesticides in wastewater. It is an HPLC method, capable of detecting about 21 pesticides of the carbamate and urea classes. It is limited in the sense that it cannot adequately resolve two pairs; monuron and monuron–TCA, and fenuron and fenuron–TCA. Additional qualitative technique, such as GC–MS, should be adopted when analysing an unfamiliar sample. Quantitation is often done using a UV detector, and, to reduce the level of interference in the analysis, Florisil clean-up is recommended [59].

Ref. [60] developed a method for detecting sixteen pesticides having different physical and chemical properties. In this method, SPME (solid-phase microextraction) was coupled with GC–MS (gas chromatography–mass spectrometry); it was deemed suitable and fitting for the study, as the range for the limits of quantification (LOQs) was 0.05–0.5 μg/L. The specific recoveries were in the percentages 75 to 140 and uncertainties of real sample concentration were less than 10%. In another study, single-drop microextraction (SDME) followed by GC–MS was used to determine four pesticides, including parathion and ethion in water samples. Recovery tests were done to evaluate its reliability and they were in the percentage range of 76.2 to 107%. The LOD and LOQ for all the pesticides of interest ranged from 0.05–0.38 μg/L and 0.15–1.1 μg/L, respectively. These results proved this method to be highly sensitive and able to detect and quantify low pesticide concentrations in water [61].

In the same vein, HPLC was also used for the quantitative analysis of organophosphate and carbamate pesticides and there was a good separation, with detection of 100 ng for all CM and OP pesticides [62]. A method for the determination of CM and OP pesticides in carrot, tomatoes, cabbage, and apples using liquid chromatography–mass spectrometry was developed. SPE was used for the extraction, the obtained recoveries obtained were in the range of 70–110%, with a few exceptions, and the relative standard deviations were less than 8% [63]. Moreover, Ref. [64] used SDME (single-drop micro-extraction) coupled with GC (gas chromatography) for the analysis of CM and OP pesticides in water; the LODs obtained were as low as between 0.02 and 0.50 ng/mL with good linearity in the range of 0.5 to 200 ng/mL. 

Thin-layer chromatography (TLC) was used for the detection of chlorinated hydrocarbons, organophosphates, and other contaminating pesticides in commercial pesticide formulations in a study by [65] where chloroform was used for the extraction of the samples that were spotted on thin-layer plates and compared with standards. Gas chromatography and infrared techniques were used for identification where the R_f_ (retention factor) for organophosphates ranged from 11 (malathion) to 94 (trithion). In a study by [66] three enzymes, RLE, BS2, and CUT (rabbit liver esterase, Bacillus subtilis esterase, and cutinase from Fusarium solani pisi) were used to detect 21 organophosphates and carbamate pesticides with HPTLC–EI (high-performance thin-layer chromatography–enzyme-inhibition assays). In this study it was found that RLE was inhibited by all the studied insecticides except acephate and recorded best results in terms of sensitivity when compared with BS2 and CUT. In another study, capillary electrophoresis (CE) with solid phase microextraction (SPME) was used to simultaneously determine seven carbamate pesticide residues in vegetables. In this study, a phosphate buffer solution (PBS) containing 45 mmol/L NaCl and 25 mmol/L cyclodextrin was used as a separation solution for CE and the LODs of the pesticides ranged from 0.1 to 0.5 μg/L; it was concluded that the method was a rapid and accurate one for the determination of these pesticides because of its high sensitivity, good reproducibility, and wide linear range [67].

### 7.4. Fluorescence Techniques

Carbamate pesticides were assessed rapidly using a fluorescence method that was developed. The calibration curve for methomyl ranged as 0.017–0.5 µg/mL with a limit of detection (LOD) of µg/mL, and this was superior to the results obtained using HPLC coupled with UV detection method [68] A pH-sensitive fluorescence probe was synthesized to determine CM and OP pesticides based on their inhibition of AChE (acetylcholine esterase), and the inhibition percentage of the activity of the enzyme was associated with the pesticide concentration [69] Fluorescence spectrometry was used to determine residues of methomyl in vegetables. The regression equation of the standard curve obtained for the method was y = 91.5242 × −0.6143, and the correlation coefficient was R^2^ = 0.9970 [70]. A method that was derived from techniques that are fluorescence-based was developed by using core–shell quantum dots (QDs) for detecting three carbamates: methomyl, carbofuran, and aldicarb in medicinal plants. The detected concentrations were as low as 0.05 μM and the detection sensitivities were high for the studied carbamates. The molecular docking study showed that these carbamates are bound to the active catalytic site of AChE through H–π and π–π interactions. That revealed the potential mechanism of the differences in strength inhibition between these pesticides on AChE [71]. 

Fluorescence based technique, using core–shell quantum dots (QDs) to detect three carbamates (methomyl, aldicarb, and carbofuran) in medicinal plants was developed by [71], where the lowest concentration detected was no more than 0.05 µM when the optimal experimental conditions were investigated. 

### 7.5. Spectrometric Techniques

A modified QuEChERS method was combined with cloud point extraction prior to spectrophotometric analysis, as described by Karnsaard et al. (2013) described to determine carbaryl residues in vegetable samples. The LOD obtained in this method was 0.1 mg/kg, which is ten times lower than analysis with no pre-concentration. Recoveries > 79% and precision with the relative standard deviation less than 11% were achieved. These were in good agreement with the results obtained from high-performance liquid chromatography [51]. Eighteen carbamates were simultaneously determined in soil using modified QuEChERS combined with liquid chromatography–tandem mass spectrometry. DCM (dichloromethane) and acetone were used for the extraction of the target pesticides in soil, then dispersive solid-phase extraction was used for the clean-up of the extracts and LC–MS–MS was used for the analysis. All the 18 target pesticides had correlation coefficients higher than 0.995. Samples spiked recoveries at 1 μg/kg and 10 μg/kg ranged from 64.9–94.2% and 64.7–104.7%, respectively, with relative standard differences in the range of 1.98–16.83%. The LODs were from 0.010 to 0.130 μg/kg. This method was deemed simple, sensitive, and inexpensive [72]. Table 8 below shows the comparison of various analytical methods for organophosphate and carbamate pesticides.

## 8. Levels of OP and CM Pesticides in the Environment

Data monitoring worldwide is poor, and more so in developing countries. This is because it is relatively expensive to sample at crucial times of the year, and the analysis of organic chemicals requires adequate facilities, which can be costly [81]. The use of pesticides globally has increased by 46% over 20 years from 1996, according to the Food and Agriculture Organization Corporate Statistical (FAOSTAT) database. Due to this, the minimization of the harmful effects of pesticides on the environment and human health is imperative, and that can be achieved by adopting regulation practices together with management and proper use of pesticides [82]).

The concentrations of CM and OP pesticides in different matrices around the world are presented in Table 9. The reported levels in the sediment matrix ranged from 800 × 10^−6^ μg/kg to 51 μg/kg, with the lowest concentration recorded in KwaZulu-Natal (Ubombo and Ingwavuma districts) [83] and the highest concentration recorded in northwest Bangladesh [84]. The high concentrations recorded in Bangladesh can be attributed to the fact that agricultural activities from neighboring areas polluted the sampling site. The lowest concentration for water matrix (0.032 μg/L) was recorded in the Gallion river in France [85], while the highest concentration of 9000 μg/L was recorded in North America [86]; high-performance liquid chromatography was used in both of these studies. The concentrations in vegetables and fruits were also reported, with leafy vegetables from Shanghai, China having the highest concentration at 22.20 μg/kg [87] and maize from Ejura, Ghana having a low concentration of 2.0 × 10^−9^ μg/kg [88].

## 9. Conclusions

Carbamate and organophosphate pesticides are the most commonly used pesticides worldwide because they are the least persistent in the environment. The occurrence and distribution of these pesticides in water and sediments are determined by their application rates, types, and physicochemical properties. OP and CM pesticides are toxic and may enter the environment through intentional release, indirect runoff, or drift. There are many extraction and analytical methods available to analyse these contaminants, including HPLC and GC, which are commonly used. Depending on their chemical structures, pesticides have different levels of toxicity. CMs can be spontaneously cleared within 48 h of exposure, while OPs can bind irreversibly to cholinesterase. The sensitive, rapid, and reliable determination of these compounds in environmental samples is important for protecting the environment and human health. CMs and OPs are hazardous to the health of humans because they inhibit the activity of acetylcholine esterase (AChE), which can lead to the malfunction of respiration and the muscle tissue of the heart and, eventually, death.

## Figures and Tables

**Figure 1 molecules-27-00618-f001:**
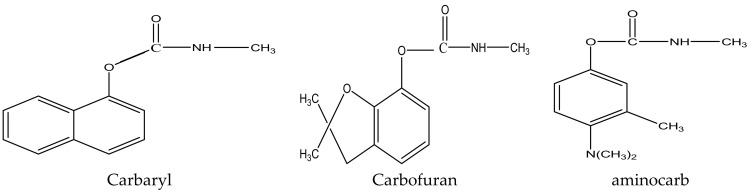
Structures of commonly used carbamate pesticides.

**Figure 2 molecules-27-00618-f002:**
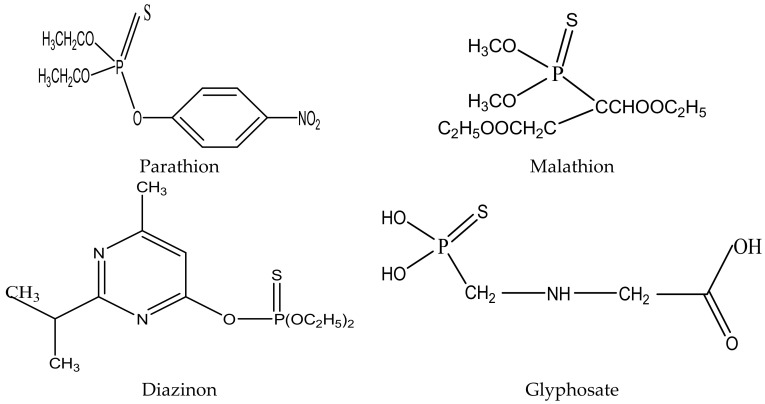
Structures of commonly used organophosphate pesticides.

**Figure 3 molecules-27-00618-f003:**
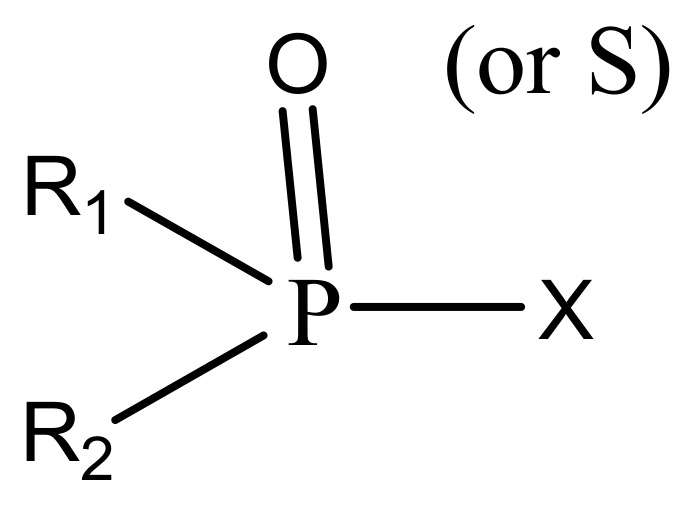
General structure of organophosphates.

**Figure 4 molecules-27-00618-f004:**
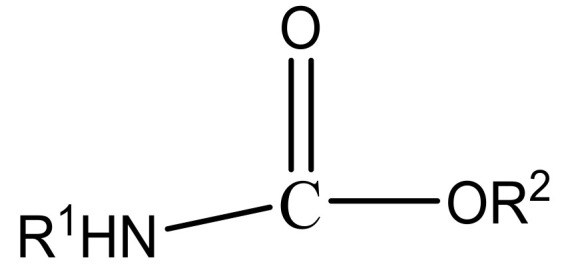
General structure of carbamate pesticides [6,14,24].

**Figure 5 molecules-27-00618-f005:**
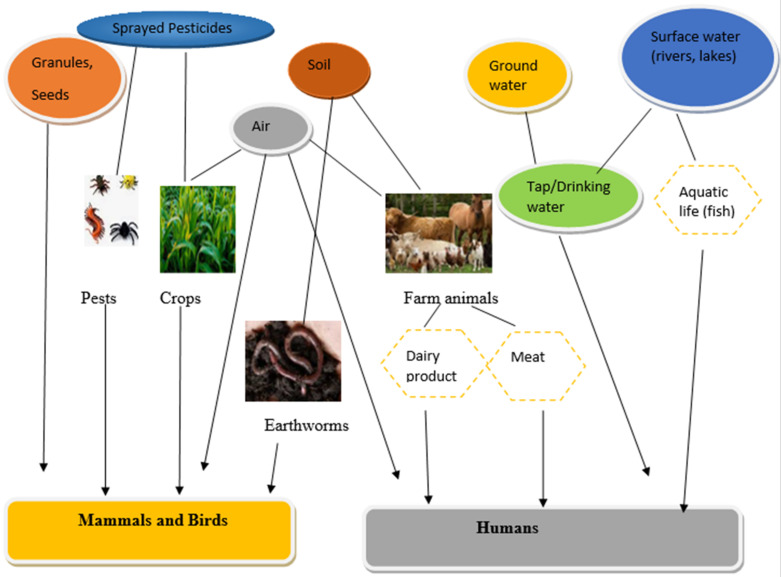
Schematic diagram of the routes of exposure of carbamate and organophosphate pesticides to wildlife and humans [34,35].

**Table 1 molecules-27-00618-t001:** Pesticides and their target organisms [5,11].

Target Pest/Organism	Type of Pesticide
larvae plants egg of insect/mites	larvicides herbicides ovicides
insects	insecticides
bacteria	bactericides
virus	virucides
ticks, mites	miticides, acaricides
molluscs	molluscicides
rodents	rodenticides
algae	algicides
fungi	fungicides
bird pests	avicides

**Table 2 molecules-27-00618-t002:** Estimation of percentage losses of major crops due to pests per year [5,13].

Crop	Percentage of Estimated Losses			
	Weeds	Diseases	Insects	Total
rice	-	-	37	37
maize	-	-	31	31
wheat	9.8	9.1	5.0	23.9
millet	17.8	10.6	9.6	38.0
potatoes	-	-	40	40
cassava	9.2	16.6	7.7	33.5
soybeans	13.5	11.1	4.5	29.1
peanuts	11.8	11.3	17.1	40.4
sugarcane	25.1	10.7	9.2	45.0

**Table 3 molecules-27-00618-t003:** Physicochemical properties of organophosphate pesticides [19,20,21].

Pesticide	Koc (cm^3^/g)	Solubility (20–25 °C) (mg/L)	V_p_ (Pa) (20–25 °C)	Half-Life T_1/2_ (Days)	Log K_ow_
acephate	0.88	650	2.26 × 10^−4^ 1.7 × 10^−6^ (23–25 °C)	13	−1.87
azinphos-methyl	1465	44	1.8 × 10^−4^	52	2.7
chlorfenvinphos	-	145	1.0 × 10^−3^	-	3.8
chlorpyriphos	-	1.4	2.7 × 10^−3^	94	4.96
diazinon	4981	60	1.2 × 10^−2^	23	3.3
dichlorvos	272	18,000	2.1	-	1.9
dimethoate	20	23	1.1 × 10^−3^	7	0.7
ethyl-parathion	5000	11	8.9 × 10^−4^	14	3.83
fenamiphos	267	700	0.12 × 10^−3^	16	3.3
fenitrothion	-	30	18 × 10^−3^	-	-
fenthion	15,000	4.2	7.4 × 10^−4^	34	4.84
malathion	1800	145	5.3 × 10^−3^	1	2.75
methamidophos	1.7	90,000	2.3 × 10^−3^	˃2.6	0.8
mevinphos	44	Miscible	1.7 × 10^−3^	3	0.13
monocrotophos	1	Miscible	2.9 × 10^−4^	30	−0.22
parathion-methyl	236	55	0.2 × 10^−3^	18.5	3.0
phorate	1000	50	8.5 × 10^−3^	60	3.9
pirimiphos-methyl	1000	9.9	2.0 × 10^−3^	10	10
terbufos	500	4.5	3.46 × 10^−2^	5	5
triazophos	-	30	0.39 × 10^−3^	-	3.3
trichlorfon	29	120,000	2.1 × 10^−4^	29	0.43

V_p_ = vapour pressure.

**Table 4 molecules-27-00618-t004:** The Half-life of some organophosphates [19,22].

Chemical	Half-Life, h
Malathion Dursban parathion	24 2256 43.0
dicapthon	6.4
dichlorofenthion	19
leptophos	48
ronnel	10.5
fenitronthion	11.2

**Table 5 molecules-27-00618-t005:** Relationship between pesticidal activity and chemical structure of CMs [6,25].

Pesticidal Activity	Common or Other Names	Chemical Structure
Herbicide	barban, chlorbufam, desmedipham, phenmedipham, swep, carbetamide, dichlormate, Asulam, karbutilate, terbucarb	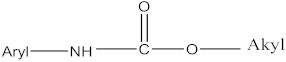 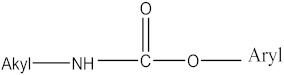
herbicides and sprout inhibitors	Chlorpropham	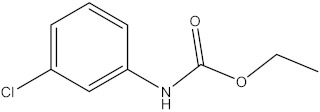
	propham	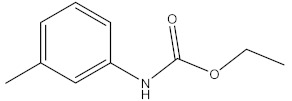
Fungicide	Benomyl, thiophanate-methyl, thiophanate ethyl, carbendazim	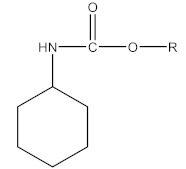
Insecticide	aldoxycarb, aminocarb, BPMC, bendiocarb, butacarb, carbanolate, carbaryl, bufencarb, carbofuran, cloethocarb, dimetilan, methiocarb	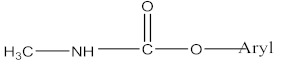 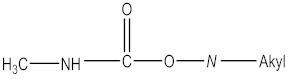
	aldicarb	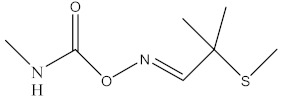

**Table 6 molecules-27-00618-t006:** Summary of physicochemical properties of some carbamates [6,27].

Name	EPA Toxicity Classification	Kow	MW	Koc	Water Solubility	V_p_
Bendiocarb/ ficam	Class II	50	223.23 g/mol	570	40 mg/L at 20 °C 260 mg/L at 25 °C	5 × 10^−6^ mm Hg at 25 °C
methomyl	Class I	3.98	162.210 g/mol	51.72, 160	10 g/L at 25 °C	5.0 × 10^−5^ mm Hg at 25 °C
aprocarb/ propoxur	Class II for oral exposures and Class III for dermal and inhalation exposures	1.4	209.245 g/mol	30	1750 mg/L at 25 °C	3 × 10^−6^ mm Hg at 20 °C 1 × 10^−2^ mHg at 12 °C

MW = molecular weight, K_OW_ = partition coefficient, K_OC_ = soil sorption coefficient, V_p_ = vapour pressure.

**Table 7 molecules-27-00618-t007:** Comparison of various extraction techniques for pesticides in solid samples.

Extraction Technique	Cost, T, and P	Solvent Type/Solvent Consumption/Extraction Time	Advantages	Disadvantages	References
Soxhlet	low cost boiling point of solvent atm. pressure	organic solvent 60–500 mL 6–24 h	It does not require filtration; samples in large amounts; easy to operate; does not depend on the matrix	Extraction time is long; large consumption of solvents; sample must be preconcentrated after extraction	[47,48]
supercritical fluid extraction (SFE)	high cost 70–150 °C 15–50 MPa	CO_2_ 10–40 mL 30–60 min	Friendly to the environment because it is not toxic; extraction is fast; uses little solvent; does not require filtration	Sample size limited; dependent on the matrix and analyte	[49]
ultrasonic-assisted extraction (UAE)	low cost 30–35 °C Atm. pressure	organic solvent 30–100 mL 30–60 min	Fast and easy to operate; large amount of sample; does not depend on the matrix	Risk of being exposed to the solvent vapour; large amount of solvent, labour intensive; requires filter	[45,46]
microwave-assisted extraction (MAE)	moderate cost 100–150 °C Atm. pressure	organic solvent 10–40 mL 20–30 min	Uses small solvent and is fast full control of extraction parameters	Filtration required; solvent must be polar; exhaustive extraction	[48]
Pressurized liquid extraction (PLE)	high cost 100–150 °C 7–15 MPa	organic solvent 10–60 mL 10–60 min	Uses small solvent and is fast; does not require filtration and is easy to use	Extraction efficiency dependent on matrix	[45,50]
subcritical water extraction (SWE)	moderate cost 200–300 °C 5 MPa	water 30–60 mL 30–60 min	Uses water, which is non-toxic, fast, friendly to the environment; uses little solvent	Optimization of operating conditions required	[45]

T = temperature, P = pressure, atm. pressure = atmospheric pressure.

**Table 8 molecules-27-00618-t008:** Comparison of various analytical methods for OP and CM pesticides.

Analytical Methods	Advantages	Disadvantages	References
electrochemical	Quick and simple measurements Good detection limits Easy sample preparation Small amount of sample (up to 50 μL using screen printed electrodes)	Total reducing power Not selective to a family of molecules unless the electrode is modified	[73]
surface-enhanced Raman Spectroscopy (SERS)	High sensitivity, simple and rapid, label free	Lack of active substrates, poor portability, poor reproducibility, limitations on batch fabrication, high cost	[74]
solid-phase microextraction (SPME)	Allows attainment of satisfactory LODs and cleaner chromatograms for volatile analytes SPME in combination with GC/MS or LC is a solvent-free or almost solvent-free procedure, obviating the need for further preparation steps	SPME fibres are not uniformly sensitive to all compounds	[75,76]
GC–MS	Very good recovery value Sensitive method	Not capable of directly analysing compounds that are nonvolatile, polar, or thermally labile	[77,78]
GC–µECD	Very good for determination of organophosphorus pesticides Highly sensitive Low detection limit	Only volatile compounds can be analysed	[79]
thin-layer chromatography (TLC)	Equipment needed is inexpensive Convenient and simple to use Consumes smaller amounts of solvents	Preparative applications are limited. Oxidation may occur if the TLC plate is stored for a while since a large surface is exposed to atmospheric oxygen	[80]
high-performance liquid chromatography (HPLC)	High quality separations are achievable Coupling with MS is well established	More time-consuming and expensive	[80]

**Table 9 molecules-27-00618-t009:** Reported concentrations of CM and OP pesticides in different matrices around the world.

Sample Source	Matrices	Concentrations Reported	Analytical Method	References
Martinique Island in the French West Indies	sediments water	44 µg/kg (chlordecone) 0.083 µg/L (aldicarb sulfone)	HPLC	[85]
Pakistan, Indus River	sediments	0.069 ± 0.0023 μg/g WW (carbofuran)	HPLC	[89]
northwest Bangladesh	water sediments	chlorpyrifos 9.1 μg/L 51 μg/kg	GC–MS	[84]
Shanghai China	leafy vegetables	22.20 μg/kg	GC/FTD	[87]
North America	water	9000 μg/L		[86]
Capot River in France	water	0.043 and 0.052 μg/L	HPLC	[85]
Galion River in France		0.083 and 0.032 μg/L		
Mekong Delta, Vietnam	surface water soils and sediments	(fenobucarb) 0.11 μg/L 1.7 and 4.3 μg/kg		[90]
Botswana (Africa)	cabbage	methamidophos 0.0262 mg/kg Methomyl 0.0140 mg/kg	LC–MS/MS	[91]
Lagos, Nigeria	sorghum and beans fruits and vegetables	Dichlorvos 2.00 ng/g Chlorpyrifos 0.002 and 60 ng/g Methiocarb 30 ng/g and 70 ng/g	GC–MS	[92]
Ejura, Ghana	maize cowpea	organophosphates 0.002–0.019 mg/kg 0.002–0.015 mg/kg	GC–ECD GC–PFPD	[88]
Zhejiang, China	soil	Parathion 43.3 ng/g	GC–MS	[93]
Lebanon	ground water drinking water ground water surface water rain water	diazinon 4.2 ng/L 2.2 ng/L 7.49 ng/L 15.8 ng/L	GC–MS	[94]
Indus River, Punjab, Pakistan,	channa Marulius muscles sediments	carbofuran 0.613–0.946 μg/g 0.069–0.081 μg/g	HPLC	[89]
Jamaica	maternal urine samples	diethylphosphate 29.0 μg/L	GC–MS	[95]
Shanghai, China	Fuji apples	carbaryl 0.5 μg/g	GC–MS	[96]
Bangladesh	water	diazinon 0.9 μg/L carbofuran 198.7 μg/L	HPLC	[97]
India (Western Ghats)	fejervarya limnocharis	carbaryl 50 µg/L malathion 500 µg/L	HPLC	[98]
Hungary	water	10–100 ng/L	GC–MS HPLC	[99]
South Litani region in South Lebanon	ground water	pirimiphos-methyl 300.87 ng/L	GC–MS	[100]
KwaZulu–Natal (Ubombo and Ingwavuma districts)	sediment water sediment water sediment water	carbaryl 0.0010 μg/Kg 0.30 μg/L carbofuran 800 × 10^−6^ μg/Kg 250 × 10^−3^ μg/L carbosulfan 300 × 10^−6^ μg/kg 80 × 10^−3^ μg/L	GC–NPD GC–FID	[83]

## Data Availability

Not applicable.

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
