# Peer review of "Analytical Evaluation of Carbamate and Organophosphate Pesticides in Human and Environmental Matrices: A Review"

_molecules, 2022, doi:10.3390/molecules27030618_

Round 1

Reviewer 1 Report

In my opinion the subject of tis paper is interesting but before acceptances it needs serious correction.

  1. All structure must be rewritten. Use a specialized software. Use the same fonts for all.
  2. SubChapter 2.2. must be removed. It is not appropriate. The subject is different.
  3. References are not correct cited in the text. You mixed citation modes.
  4. There are grammatical errors and typos. All manuscript must be checked carefully.

Reviewer 2 Report

The review paper is well written and cover all important issues concerning Carbamate and Organophosphate Pesticides. The impact of the use of pesticides is highlighted and the methodology of the present manuscript content is appropriate.

Reviewer 3 Report

Reviewer report on manuscript Molecules-1518711

The submitted review article describes various aspects about the carbamate and organophosphate pesticides in human and environmental matrices.

I read the article carefully and I cannot say that I enjoyed it. The scope and the novelty of the submitted paper is not clearly documented. The majority of the data is devoted on the description of some physicochemical properties of the certain pesticides and the information about the analytical methods are very limited. To my opinion, the article is unbalanced and its quality is very low for the Journal’s standards.

Based on the above statements, I recommend rejection of the submitted paper.

Round 2

Reviewer 1 Report

The paper can be accepted.

Reviewer 3 Report

The manuscript has been adequately revised according to my previous comments.

Comments

1) please check the linearity concentration range in line 403.